# Improving provider-initiated testing for HIV and other STI in the primary care setting in Amsterdam, the Netherlands: Results from a multifaceted, educational intervention programme

Saskia Bogers[1,2,3]*, Maarten Schim van der Loeff[1,2,4], Anders Boyd[2,4,5], Nynke van Dijk[6,7], Suzanne Geerlings[1,2,3], Jan van Bergen[6,8], on behalf of the HIV Transmission Elimination AMsterdam (H-TEAM) Consortium[¶]

1 Amsterdam UMC Location University of Amsterdam, Internal Medicine, Amsterdam, The Netherlands, 2 Amsterdam Institute for Infection and Immunity, Infectious Diseases, Amsterdam, The Netherlands, 3 Amsterdam Public Health Research Institute, Quality of Care, Amsterdam, The Netherlands, 4 Department of Infectious Diseases, Public Health Service of Amsterdam, Amsterdam, The Netherlands, 5 Stichting HIV Monitoring, Amsterdam, The Netherlands, 6 Department of General Practice, Amsterdam UMC Location University of Amsterdam, Amsterdam, The Netherlands, 7 Center of Expertise Urban Vitality, Faculty of Health, Amsterdam University of Applied Sciences, Amsterdam, The Netherlands, 8 STI AIDS Netherlands, Amsterdam, The Netherlands

¶ Membership of the HIV Transmission Elimination AMsterdam (H-TEAM) Consortium is provided in the Acknowledgments.
* s.j.bogers@amsterdamumc.nl

## Abstract

### Background

In the Netherlands, general practitioners (GPs) play a key role in HIV testing. However, the proportion of people diagnosed with late-stage HIV remains high, and opportunities for earlier diagnosis are being missed. We implemented an educational intervention to improve HIV and STI testing in primary care in Amsterdam, the Netherlands.

### Methods

GPs were invited to participate in an educational program between 2015 and 2020, which included repeat sessions using audit and feedback and quality improvement plans. Data on HIV, chlamydia and gonorrhoea testing by GPs were collected from 2011 through 2020. The primary outcome was HIV testing frequency, which was compared between GPs before and after participation using Poisson regression. Secondary outcomes were chlamydia and gonorrhoea testing frequencies, and positive test proportions. Additional analyses stratified by patient sex and age were done.

### Findings

GPs after participation performed 7% more HIV tests compared to GPs before participation (adjusted relative ratio [aRR] 1.07, 95%CI 1.04–1.09); there was no change in the proportion

**Data Availability Statement:** All relevant data are within the paper and its Supporting Information files.

**Funding:** This study was funded by Aidsfonds (grant number: P-42702; funding acquired by SEG. www.aidsfonds.nl) and the HIV Transmission Elimination Amsterdam (H-TEAM) Consortium (funding acquired by SEG. www.hteam.nl). The funders had no role in study design, data collection and analysis, decision to publish, or preparation of the manuscript.

**Competing interests:** The authors declare that they have no competing interests related to this work.

HIV positive tests (aRR 0.87, 95%CI 0.63–1.19). HIV testing increased most among patients who were female and ≤19 or 50–64 years old. After participation, HIV testing continued to increase (aRR 1.02 per quarter, 95%CI 1.01–1.02). Chlamydia testing by GPs after participation increased by 6% (aRR 1.06, 95%CI 1.05–1.08), while gonorrhoea testing decreased by 2% (aRR 0.98, 95%CI 0.97–0.99). We observed increases specifically in extragenital chlamydia and gonorrhoea testing.

## Conclusions

The intervention was associated with a modest increase in HIV testing among GPs after participation, while the proportion positive HIV tests remained stable. Our results suggest that the intervention yielded a sustained effect.

## Introduction

Globally, the annual number of new HIV infections has been reduced by 52% since its peak in 1997, but an estimated 1.5 million new HIV infections still occurred in 2021 [1]. In the Netherlands, the number of newly-diagnosed HIV infections has declined by 53% since 2015, with 427 newly-diagnosed HIV infections in 2021 [2]. The Netherlands has thus reached one of their goals set by the national action plan on sexually transmitted infections (STI), HIV and sexual health: to achieve a 50% reduction in the annual number of newly-diagnosed HIV infections by 2022, compared with 2015 [3]. However, an estimated 6% of people living with HIV (PLHIV) in the Netherlands in 2021 were unaware of their diagnosis, and over half of individuals newly diagnosed were at a late-stage of HIV infection, defined as having a CD4 count below 350 cells/mm$^3$ or an AIDS-defining event [2]. Since studies have shown that the majority of HIV transmissions come from persons with undiagnosed HIV and adequate treatment of HIV prevents onward transmission, reduction of the proportion undiagnosed and timely diagnosis of HIV are crucial in ending the HIV epidemic [4–6].

In the Netherlands, general practitioners (GPs) provide the majority of sexual health consultations (71%) [7]. GPs may therefore play a key role in diagnosing HIV. Additionally, GPs may be the first healthcare provider to recognize symptoms indicating acute HIV infection, as well as HIV indicator conditions [8, 9]. Since 2019, 32% of PLHIV were diagnosed by GPs, while 28% were diagnosed at sexual health centres (SHCs) and 35% in the hospital setting. The proportions of PLHIV that were diagnosed at a late stage of infection in these settings were 46%, 30% and 81%, respectively [2]. While SHCs provide routine HIV testing for key populations on an opt-out basis, HIV diagnoses in hospitals are generally made among patients presenting with HIV indicator conditions or AIDS-defining illnesses, usually after referral by the GP. Thus, it is likely that GPs can facilitate earlier diagnosis by applying optimal HIV testing strategies. However, it has been previously shown that there were missed opportunities for earlier HIV diagnosis in the primary care setting, and that barriers and HIV related stigma hampering proactive HIV testing by GPs may delay HIV diagnosis [9–11].

The Dutch HIV epidemic is mostly concentrated in urban areas, with 26% of PLHIV, and an estimated 12% of undiagnosed PLHIV living in the city of Amsterdam [2]. In response to this epidemiological context, the HIV Transmission Elimination in Amsterdam (H-TEAM) consortium was founded in 2014 to deliver a multifaceted city-based approach to end the HIV-epidemic [12]. One of the H-TEAM's objectives is facilitating timely and frequent HIV

testing in primary care. To achieve this goal, the H-TEAM implemented a multifaceted educational intervention programme for GPs in Amsterdam as part of their efforts to improve HIV testing in primary care. To extend the impact of the intervention on quality of sexual health consultations and to make participation more rewarding for GPs, the educational programme additionally focused on other STI, including chlamydia and gonorrhoea. In this study, we evaluated the effect of the educational intervention on HIV, chlamydia and gonorrhoea testing by GPs in Amsterdam, the Netherlands.

## Material and methods

### Setting and participants

All Amsterdam GPs were invited to participate in an educational intervention by a partner organisation that facilitates integrated healthcare services in primary care. The educational sessions were delivered between February 2015 and December 2020, when saturation in interest to participate was achieved (i.e. no more GPs were interested to participate). The sessions were attended by practicing groups of GPs attending continuing medical education (CME) sessions together. One GP from each group attended a teach-the-teacher session for the programme and moderated the sessions. GPs received points for participation, which are needed to remain accredited.

### Intervention

The educational intervention was designed by CME coordinators in conjunction with experts in the field of sexual health and HIV/STI. This intervention used evidence-based elements for effective interventions, including interactive audit and feedback, multiple exposures, and small-group sessions [13]. The programme consisted of two consecutive educational sessions. During the first session, several topics on sexual health and appropriate HIV and STI testing strategies were discussed, including indications for extragenital chlamydia and gonorrhoea testing, and testing for HIV in the case of HIV indicator conditions or symptoms associated with acute HIV infection. A member of the national expert group on HIV and STI in primary care provided updates on state-of-the-art HIV and STI testing and care. Interactive graphical audit and feedback was then given to the participants on their HIV, chlamydia and gonorrhoea testing frequency and positivity, compared to the city average [13]. At the end of the first session, each group established quality improvement plans for optimal HIV and STI testing and care in their practice. During the second session, the implementation of these quality improvement plans was evaluated, updates on HIV and STI epidemiology, diagnosis, prevention and treatment were provided, and updated graphical audit and feedback was given to participants. During both sessions, participants received educational materials including workbooks, STI testing flowcharts, information flyers, and further reading materials. Finally, in the context of this educational intervention programme, GPs received digital newsletters several times a year with relevant news updates on HIV and STI testing and care. More details on the educational programme are described elsewhere [13].

### Data collection

Laboratory data on HIV, chlamydia and gonorrhoea tests ordered by all GPs in Amsterdam from 2011 through 2020 were collected from seven major diagnostic laboratories for primary care using a standardized data request form. These data were used to generate the graphical audit and feedback for each session and to evaluate the effect of the intervention. Participating laboratories provided data on the ordering GP (i.e., 4-digit postal code of their practice), test

ordered (i.e., date, anatomical site of sampling, and test result), and the patient who was tested (i.e., age and sex). During the educational sessions, we assessed which laboratories the participating GPs were using through a questionnaire. Based on the responses, we estimated that 90–95% of all HIV, chlamydia and gonorrhoea tests ordered by GPs in Amsterdam were included in the data provided by these laboratories.

## Outcomes

The primary outcome was the number of HIV tests ordered per GP per quarter. Secondary outcomes were the number of HIV tests that were positive, the overall number of chlamydia and gonorrhoea tests ordered per GP per quarter, the number of urogenital, anorectal and oropharyngeal chlamydia and gonorrhoea tests ordered per GP per quarter, and their respective proportions positive.

## Statistical analysis

Overall trends in HIV, chlamydia and gonorrhoea testing over time were calculated per 10,000 residents of Amsterdam. We modelled outcomes by quarter-year periods using Poisson regression. Each record represented a quarter-year period of one GP. A record could concern (1) a GP prior to participation, (2) a GP after participation, or (3) a GP who never participated. The model was used to estimate the relative ratio (RR) and its 95% confidence interval (CI) of the mean number of tests or proportion positive comparing between (2) GPs after participation and (3) GPs who never participated in the intervention with GPs before participation (1; reference group). For participants, time after participation started on the date a GP first attended a session in the programme, regardless of whether they attended one or both sessions. Due to the disruption of healthcare service delivery from COVID-19, data from quarters 2–4 in 2020 were excluded from analysis, and follow-up therefore ended on March 31st, 2020. We added city district of the ordering GP (as the HIV and STI incidence and prevalence vary by district) and year of testing (to correct for any secular trends) as covariates to the model. For the primary outcome, a sensitivity analysis was performed by excluding GPs who ordered >30 HIV tests per quarter-year before participation (among participants) or before the start of the programme (among GPs who never participated), as both participation and effect of the intervention were expected to be low since these GPs already had high levels of HIV testing activity. Additional analyses using the same models stratified by patient sex and age categories (<20 years, 20–34 years, 35–49 years, 50–64 years and ≥65 years) were performed. Finally, we estimated the effect of the intervention over time by regressing the outcomes by GPs after participation on quarter-years, adjusted for city district and year of testing. A p-value of <0.05 was considered statistically significant. Data analysis was performed using Stata (v15.1, College Station, Texas, USA).

## Ethics statement

All GPs in Amsterdam were provided with the opportunity to object to use of their laboratory data through a written opt-out procedure. All participating GPs provided written informed consent for the use of the results of the educational sessions and their evaluations for research purposes. The Medical Ethics Committee of the Amsterdam University Medical Centres, University of Amsterdam determined that this study does not meet the definition of medical research involving human subjects under Dutch law (file W18_230, #18.274, 24 July 2018).

## Results

### Participation in the programme

The mean annual number of registered GPs practicing in Amsterdam in 2011–2018 was 504 [14]. In total, 36 first and second educational sessions were conducted, with 229 unique GPs attending. A third (75/229) of GPs attended both sessions of the programme. First sessions were conducted between February 2015 and April 2019 and second sessions were conducted between November 2017 and December 2020.

### Data collected

Data on 106,424 HIV tests, 343,648 chlamydia tests and 321,345 gonorrhoea tests by Amsterdam GPs from January 2011 through March 2020 were collected. Of these, 684 HIV tests, 24,318 chlamydia tests and 6,984 gonorrhoea tests were positive, resulting in 0.6%, 7.1% and 2.2% positive tests, respectively. Overall, the data collected during the study period concerned tests ordered by 725 GPs, with a mean of 464 GPs per year (i.e. 464/504; 92% of the mean number of registered GPs).

### HIV testing

From 2011–2014, overall HIV testing by GPs decreased with 34%, from 175 to 116 per 10,000 residents of Amsterdam. From 2015 onward, overall HIV testing increased by 10%, from 123 to 135 per 10,000 residents in 2020 (**Fig 1**).

The overall median number of HIV tests ordered per GP per quarter was 5, interquartile range (IQR) 2–9. By GP group, this was 5, IQR 2–9 for GPs before participation and 5, IQR 3–9 for GPs after participation in the intervention. From 2011 to 2020, the median number of HIV tests ordered by GPs who never participated decreased from 5, IQR 2–11 to 4, IQR 2–8. We observed a 7% increase in HIV testing among GPs after participation, compared to before participation (relative ratio [RR] adjusted for calendar year and city district 1.07, 95%CI 1.04–1.09, p<0.001), (**Table 1** and **Fig 2**).

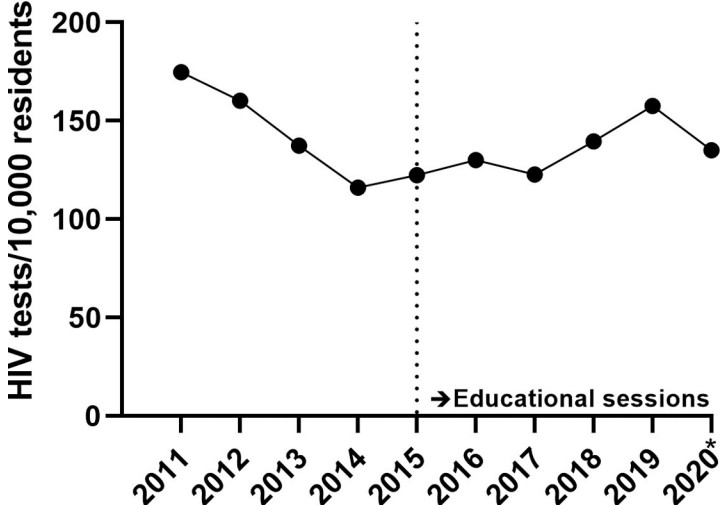

**Fig 1. Trends in HIV tests performed by GPs per 10,000 residents of Amsterdam, 2011–2020.** *2020 data only include the first quarter. The dotted vertical line represents the transition to the period in which the educational sessions were implemented.

**Table 1. Adjusted relative HIV test ratios among GPs in Amsterdam after participation in an educational intervention, compared to GPs before participation, overall and by patient sex and age, 2011–2020.**

| | Main analysis | | Sensitivity analysis* | |
|---|---|---|---|---|
| | Relative HIV Test Ratio | 95% CI | Relative HIV Test Ratio | 95% CI |
| Overall | 1.07 | 1.04–1.09 | 1.09 | 1.07–1.12 |
| **By sex** | | | | |
| Males | 1.06 | 1.03–1.10 | 1.07 | 1.03–1.10 |
| Females | 1.08 | 1.04–1.12 | 1.13 | 1.09–1.17 |
| **By age categories** | | | | |
| ≤19 years | 1.20 | 1.02–1.41 | 1.26 | 1.06–1.49 |
| 20–34 years | 1.08 | 1.04–1.12 | 1.10 | 1.06–1.14 |
| 34–49 years | 1.01 | 0.97–1.05 | 1.03 | 0.99–1.08 |
| 50–64 years | 1.17 | 1.10–1.24 | 1.23 | 1.15–1.32 |
| ≥65 years | 0.95 | 0.83–1.09 | 0.94 | 0.80–1.10 |
| **Males by age categories** | | | | |
| ≤19 years | 1.15 | 0.87–1.50 | 1.17 | 0.88–1.56 |
| 20–34 years | 1.05 | 1.00–1.10 | 1.04 | 0.99–1.10 |
| 34–49 years | 1.02 | 0.97–1.08 | 1.02 | 0.96–1.08 |
| 50–64 years | 1.17 | 1.09–1.26 | 1.23 | 1.13–1.33 |
| ≥65 years | 0.99 | 0.85–1.15 | 0.99 | 0.83–1.19 |
| **Females by age categories** | | | | |
| ≤19 years | 1.23 | 1.01–1.51 | 1.32 | 1.06–1.65 |
| 20–34 years | 1.12 | 1.07–1.18 | 1.17 | 1.12–1.24 |
| 34–49 years | 0.99 | 0.93–1.06 | 1.05 | 0.98–1.12 |
| 50–64 years | 1.15 | 1.02–1.30 | 1.21 | 1.06–1.38 |
| ≥65 years | 0.78 | 0.54–1.11 | 0.69 | 0.47–1.03 |

Relative test ratios were adjusted for city district of the ordering GP and year of testing.

*Excluding GPs that had already ordered >30 HIV tests per quarter before participation in the educational intervention.

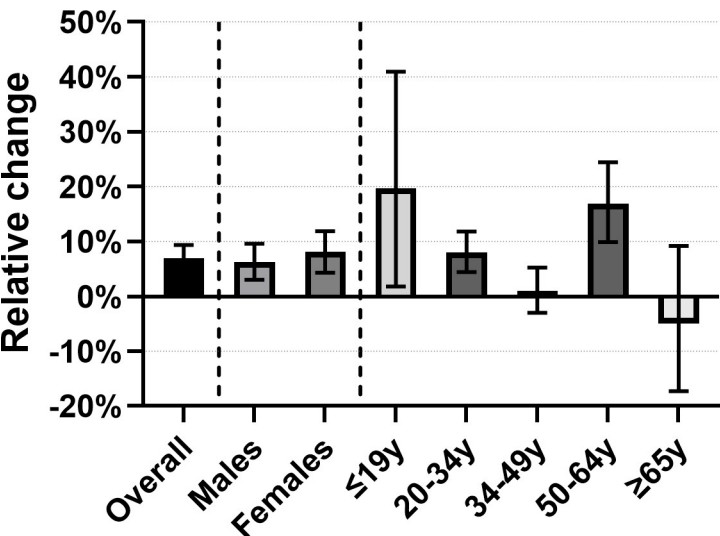

**Fig 2. Relative change and 95% confidence intervals in HIV testing among GPs in Amsterdam after participation in an educational intervention, compared to GPs before participation, overall and by patient sex and age, 2011–2020.**

By patient sex, this increase was 6% among men and 8% among women. By age categories, the largest increases in HIV testing among GPs after participation were observed in those ≤19 or 50–64 years old. Among men, significant increases in HIV testing by GPs after participation were only observed among patients aged 50–64 years old. Among women, significant increases were observed among patients aged ≤19, 20–34 and 50–64 years old (**Table 1 and Fig 2**). In sensitivity analyses excluding GPs who ordered >30 HIV tests per quarter-year at baseline (n = 9 among GPs who participated and n = 22 among GPs who never participated in the programme), we observed a 9% increase in overall HIV testing among GPs after participation in the intervention, a 7% increase among male patients, a 13% increase among female patients, and increases among all groups of patients aged <65 years old, **Table 1**.

The overall median number of HIV tests ordered was 4, IQR 2–8 for GPs who never participated in the intervention. This group ordered 18% more HIV tests compared to GPs before participation (aRR 1.18, 95%CI 1.16–1.20, p<0.001). However, in sensitivity analyses excluding GPs who ordered >30 HIV tests per quarter-year at baseline, we observed no difference in testing among GPs who never participated compared to GPs before participation (aRR 0.98, 95% CI 0.97–1.00, p = 0.07).

## HIV positivity

From 2011 to 2020, overall HIV positivity declined from 0.8% to 0.5%. By participant group, 174/25,909 (0.7%) HIV tests ordered by GPs before participation, and 72/15,509 (0.5%) tests ordered by GPs after participation in the intervention were positive; no significant change was observed in the proportion positive HIV tests ordered per quarter by GPs after participation compared to GPs before participation (aRR 0.87, 95%CI 0.63–1.19, p = 0.39). Similar results were found in analyses when stratified by patients' sex and age, **Table 2**.

## Chlamydia testing and positivity

From 2011–2014, overall chlamydia testing decreased with 18%, from 453 to 373 per 10,000 residents of Amsterdam. From 2015 onward, overall chlamydia testing increased by 44%, from 397 to 569 per 10,000 residents. The overall median number of chlamydia tests ordered per GP

**Table 2. Adjusted relative HIV positivity ratios among GPs in Amsterdam after participation in an educational intervention, compared to GPs before participation, overall and by patient sex and age, 2011–2020.**

|  | Main analysis | | Sensitivity analysis* | |
|---|---|---|---|---|
|  | Relative HIV Positivity Ratio | 95% CI | Relative HIV Positivity Ratio | 95% CI |
| Overall | 0.87 | 0.63–1.19 | 0.88 | 0.61–1.26 |
| **By sex** | | | | |
| Males | 0.82 | 0.58–1.16 | 0.90 | 0.60–1.34 |
| Females | 1.03 | 0.47–2.22 | 0.84 | 0.37–1.93 |
| **By age categories** | | | | |
| ≤19 years | n/a | n/a | n/a | n/a |
| 20–34 years | 0.76 | 0.39–1.48 | 0.60 | 0.29–1.24 |
| 34–49 years | 1.06 | 0.66–1.72 | 1.27 | 0.73–2.19 |
| 50–64 years | 0.72 | 0.40–1.30 | 0.73 | 0.37–1.46 |
| ≥65 years | n/a | n/a | n/a | n/a |

Relative test ratios were adjusted for city district of the ordering GP and year of testing.

*Excluding GPs that had already ordered >30 HIV tests per quarter before participation in the educational intervention. n/a: parameter estimates could not be obtained due to low numbers.

per quarter was 17, IQR 9–27. By GP group, this was 17, IQR 10–27 for GPs before participation and 20, IQR 11–33 for GPs after participation in the intervention. From 2011 to 2020, the median number of chlamydia tests ordered by GPs who never participated remained stable from 16, IQR 8–27 to 16, IQR 8–28. We observed a 6% increase in chlamydia testing among GPs after participation, compared to GPs before participation (aRR 1.06, 95%CI 1.05–1.08, p<0.001, **Fig 3** **and** **S1 Table**), which did not vary by patient sex.

By patient age categories, 10% and 16% increases in chlamydia testing were observed among patients aged 20–34 and 50–64 years old, respectively, while a 3% decrease was observed among 34–49 year olds. By anatomical site, we observed a 5% increase in urogenital chlamydia testing among GPs after participation, while there was a 40% increase in anorectal chlamydia testing and a 15% increase in oropharyngeal chlamydia testing. The largest increase in chlamydia testing was observed for anorectal chlamydia in women (aRR 2.10, 95% CI 1.81–2.44, p<0.001, **Fig 3** **and** **S1 Table**).

Overall, 5,668/85,611 (6.6%) chlamydia tests ordered by GPs before participation and 4,346/60,018 (7.2%) tests ordered by GPs after participation were positive. No significant change was observed in the overall proportion positive chlamydia tests ordered by GPs after participation (aRR 1.02, 95% CI 0.98–1.07, p = 0.36, **S1 Table**), but we did observe an increase in positivity among patients aged 50–64 years (aRR 1.37, 95% CI 1.10–1.70, p = 0.01). By anatomical site, no change in the overall proportion positive chlamydia tests was observed.

## Gonorrhoea testing and positivity

From 2011–2014, overall gonorrhoea testing decreased with 23%, from 456 to 349 per 10,000 residents of Amsterdam. From 2015 onward, overall gonorrhoea testing increased by 34%, from 375 to 504 per 10,000 residents. The overall median number of gonorrhoea tests ordered per GP per quarter was 15, IQR 8–25. By GP group, this was 17, IQR 10–26 for GPs before participation and 17, IQR 10–27 for GPs after participation in the intervention. From 2011 to 2020, the median number of gonorrhoea tests ordered by GPs who never participated decreased from 16, IQR 8–27 to 15, IQR 7–26. We observed a 2% decrease in gonorrhoea testing among GPs after participation, compared to GPs before participation (aRR 0.98, 95% CI 0.97–0.99, p<0.001, **Fig 3** **and** **S2 Table**). This decrease was only observed among women, while a 3% increase was observed among men. By age, 8% and 9% decreases in gonorrhoea testing were observed among patients aged ≤19 and 34–49 years old, respectively, while a 12% increase was observed among 50–64 year olds. By anatomical site, we observed a 5% decrease in urogenital gonorrhoea testing among GPs after participation, while there was a 36% increase in anorectal gonorrhoea testing, and a 9% increase in oropharyngeal gonorrhoea testing. The largest increase in gonorrhoea testing was observed for anorectal gonorrhoea in women (aRR 1.98, 95% CI 1.69–2.32, p<0.001), while the largest decrease in gonorrhoea testing was observed in ≥65 year-old patients being tested for urogenital gonorrhoea (aRR 0.80, 95% CI 0.71–0.91, p<0.001, **Fig 3** **and** **S2 Table**).

Overall, 1,347/81,974 (1.6%) gonorrhoea tests ordered by GPs before participation and 1,392/50,616 (2.8%) tests ordered by GPs after participation were positive. No significant change was observed in the overall proportion positive gonorrhoea tests results by GPs after participation, compared to GPs before participation (aRR 1.09, 95% CI 1.00–1.19, p = 0.052, **S2 Table**), but we did observe an increase in positivity among patients aged 35–49 years (aRR 1.23, 95% CI 1.04–1.46, p = 0.02). By anatomical site, no change in the overall proportion positive gonorrhoea tests was observed.

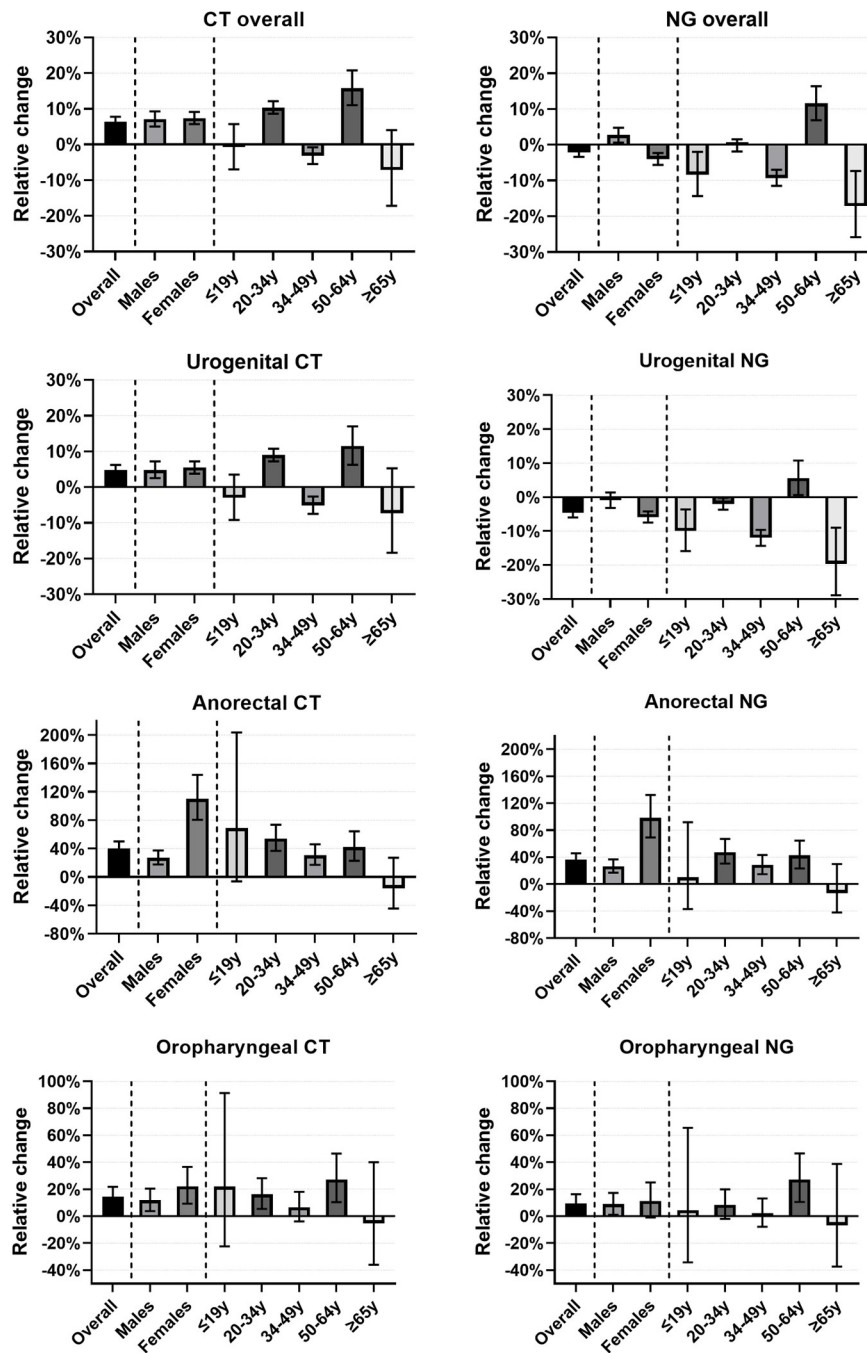

**Fig 3. Relative change and 95% confidence intervals in overall, urogenital, anorectal and oropharyngeal chlamydia and gonorrhoea testing among GPs in Amsterdam after participation in an educational intervention, compared to GPs before participation, overall and by patient sex and age, 2011–2020.** CT: Chlamydia trachomatis. NG: Neisseria gonorrhoeae. Axis ranges vary by panel.

## Trends in testing over time after participation

The median number of quarter-years of data following participation was 6, IQR 3–9 and a range of 1–19. In an analysis estimating the effect of the intervention over time among GPs after participation, we observed an increase in HIV testing over time since participation (aRR

1.02 per quarter, 95% CI 1.01–1.02, p<0.001); the same was observed for chlamydia and gonorrhoea testing overall and by anatomical site (**S3 Table**). This increase over time was largest among anorectal chlamydia tests (aRR 1.09, 95% CI 1.08–1.10, p<0.001), while it was smallest for urogenital gonorrhoea tests (aRR 1.00, 95% CI 1.00–1.01, p = 0.01, **S3 Table**).

## Discussion

We implemented an educational intervention to improve HIV and STI testing by GPs in Amsterdam. The educational intervention yielded a modest increase in the number of HIV tests ordered by GPs after participation. This increase was largest among patients who were female and those ≤19 or 50–64 years old. There was no change in the proportion positive HIV tests.

The differences in effect of the intervention by patient characteristics may suggest increased HIV testing among groups that were often overlooked previously when considering HIV testing, including women and older patients. This assertion is supported by the fact that in the Netherlands, older patients and heterosexual men and women are more commonly diagnosed at a late stage of infection compared to MSM and younger patients [2]. In 2021, the median age at HIV diagnosis was 40 years, and 30% of people newly diagnosed with HIV were 50 years or older, justifying proactive HIV testing of persons with indicator conditions and/or at risk in this age group [2]. Likewise, an increase in the percentage positive chlamydia and gonorrhoea tests done at SHCs among patients age 50 years and older was observed over the last decade [7]. Conversely, as the median age at HIV diagnosis has been increasing over time [2], the decrease in HIV tests ordered among ≥65 year-old patients may lead to missed opportunities for HIV diagnosis if this group is inadequately tested in the future, although the number of persons diagnosed with HIV at ≥65 years is small in absolute numbers. We also observed an increase in HIV testing among ≤19 year-olds, suggesting that GPs became more proactive in offering HIV tests to teenagers attending sexual health consultations. In primary care, patients <25 years old consisted of about a third of sexual health consultations and about 40% of STI diagnoses in 2019 [7]. The increase in HIV testing among ≤19 year olds may therefore have been due to an increase in adherence to the guideline for GPs on STI consultations, which recommends testing for HIV in the presence of other STI [15]. However, only 9% of new HIV diagnoses in 2020 were made among <25 year-olds, reflecting the low risk of HIV in this age group.

We observed no effect of the intervention on the proportion positive HIV tests. Given the observed increase in HIV test frequency and the strong decline in HIV incidence in the Netherlands over the last decade [2], a decline in the proportion positive test results would have been expected had the testing strategy remained the same. It is therefore likely that HIV testing became more targeted. Moreover, the percentage of positive tests observed in our study means that provider-initiated HIV testing in primary care is a cost-effective strategy, as it exceeded the previously identified cost-effectiveness threshold for routine HIV testing of 0.1% positivity [16–18].

In contrast to findings from other intervention studies that aimed to improve HIV and STI testing in primary care [19, 20], the findings from our study suggested that the effect on testing among participants did not wane over time. This may have been the result of the quality improvement plans that GPs were encouraged to make during the sessions, as well as the graphical audit and feedback, making GPs intrinsically motivated to improve their testing behaviour, as was suggested previously [21–23]. We did observe a small decrease in overall testing by GPs in Amsterdam in the first quarter of 2020, which may reflect a decrease in perceived HIV risk among patients and GPs, among other factors. This finding is of particular

importance in the context of a shrinking HIV epidemic, in which keeping GPs motivated for proactive HIV testing may be challenging in the future, when incidence and therefore perceived risk may decline.

The observed increase in the number of HIV tests ordered by GPs after participation was modest. A previous educational intervention to improve HIV testing rates in primary care in the UK showed no increase in testing which the authors ascribed to the fact that it was a single session without performance feedback, and time constraints among GPs [19]. More recently, a study using on-screen prompts to test for HIV in patients presenting with indicator conditions in Spain yielded a 3% increase (from 18% to 21%) in HIV testing rates, and the authors suggested that additional education among healthcare providers might further improve HIV testing [24]. In contrast, an intervention to improve nurse-led routine rapid HIV testing in general practice in the UK, which used a combination of training and follow-up sessions, external support, prompts and incentive payments to the practices yielded a 85% increase in testing rates. However, testing rates declined after the trial was completed [25]. These results from interventions to increase HIV testing in primary care in countries similar to the Netherlands in terms of HIV prevalence highlight the challenges of designing and implementing interventions that yield a large, sustainable increase in HIV testing in low-prevalence settings, as has been recognized by several studies that qualitatively assessed factors for success in this setting [21, 26–28].

A secondary goal of our educational intervention programme was to improve testing for other STI. We found that while overall chlamydia testing increased among GPs after participation, overall gonorrhoea testing decreased. This is in accordance with GP guidelines on STI consultations, as it recommends gonorrhoea testing only when selected risk factors are present. Therefore, the observed decrease in gonorrhoea testing may indicate closer adherence to this guideline [15]. Most notably, large increases were observed in extragenital chlamydia and gonorrhoea testing by GPs after participation; extragenital testing and the role of autoinoculation in persistent or recurrent chlamydia infections were explicitly addressed during the sessions [29]. In the past years, an increase in anorectal and oropharyngeal chlamydia and gonorrhoea diagnoses has been observed at SHCs [7]. Previous research has shown that compared to SHCs, GPs rarely ordered extragenital tests [30]. Therefore, extragenital infections are likely often being missed in primary care, particularly in women, possibly leading to suboptimal treatment. Nevertheless, the clinical relevance of asymptomatic extragenital chlamydia infections currently remains unclear [31, 32].

## Strengths and limitations

Strengths of this study include the large proportion of GPs in Amsterdam participating in the intervention, as well as the collection of comprehensive data on HIV and STI testing by nearly all Amsterdam GPs over nearly ten years. This collection allowed a more precise assessment of the intervention's impact on HIV and STI testing frequencies. Additionally, as we collected up to nearly five years of follow-up data on GPs after participation in the intervention, we were able to assess its impact over a longer period of time, thereby estimating the sustainability of the intervention's effect. We do, however, recognize several limitations of this study. Foremost, our data did not include any parameters on patient HIV and STI risk, and therefore no risk-based stratification of our outcomes could be made. Furthermore, we could not collect additional data on patients testing HIV positive, and therefore could not determine whether the proportion diagnosed at a late stage of HIV infection decreased among GPs after participation in the intervention. Such data, as well as qualitative analyses among GPs who participated, could further indicate how the quality of HIV and STI testing improved among participants,

in addition to the quantity of testing reported. While the overall participation to this pro-
gramme was good, only a third of participating GPs participated in both sessions. This may
have hampered the programme's potential, but it also indicates that GPs may have been too
constrained for time to attend a second session on this topic. Finally, while we included year of
testing in our model to correct for any secular trends, we could not correct for any other fac-
tors that may have influenced individual GP's testing behaviour. This educational intervention
was part of several H-TEAM initiatives to improve provider-initiated HIV testing in Amster-
dam. Consequently, Amsterdam GPs were exposed to multiple initiatives, including local HIV
test weeks, pre-exposure prophylaxis (PrEP) campaigns, newsletters on HIV and STI, and
media coverage of H-TEAM activities [12, 13]. We previously reported that after an initial
decline in HIV testing by GPs in Amsterdam, a stabilization in testing coincided with the start
of our intervention [13], and this trend may therefore reflect the overall effect from a multilevel
and comprehensive city-based approach.

## Conclusions

The educational intervention was associated with a significant, but modest increase in HIV
testing among GPs after participation, while the proportion positive HIV tests remained stable.
Our results suggest that the effect of the intervention was sustained over time.

## Supporting information

**S1 Table. Adjusted relative overall, urogenital, anorectal and oropharyngeal chlamydia
test ratios and positivity ratios among GPs in Amsterdam after participation in an educa-
tional intervention, compared to GPs before participation, overall and by patient sex and
age, 2011–2020.** Relative test ratios are adjusted for city district of the ordering GP and year of
testing. n/a: parameter estimates could not be obtained due to low numbers.
(DOCX)

**S2 Table. Adjusted relative overall, urogenital, anorectal and oropharyngeal gonorrhoea
test ratios and positivity ratios among GPs in Amsterdam after participation in an educa-
tional intervention, compared to GPs before participation, overall and by patient sex and
age, 2011–2020.** Relative test ratios are adjusted for city district of the ordering GP and year of
testing. n/a: parameter estimates could not be obtained due to low number.
(DOCX)

**S3 Table. Relative trends in HIV, chlamydia and gonorrhoea testing over quarter-year
periods among GPs in Amsterdam after participation in an educational intervention,
2011–2020.** *The relative test ratio indicates the number of ordered tests by a GP who partici-
pated in the intervention in one quarter, relative to the previous quarter.
(DOCX)

## Acknowledgments

We thank all general practitioners who participated in this educational intervention. We grate-
fully acknowledge all parties involved in the design and implementation of the educational
intervention programme: Nienke Brinkman for project communication and implementation,
Elaa, and Landelijke Huisartsen Vereniging for their collaboration, Michelle Kroone and Vita
Jongen for data management, AMC-LAKC, Comicro BV, OLVG Laboratory, Reinier Haga
MDC, GGD-Streeklab, ATAL-Huisarts and SHO for collaborating in data collection. We

further acknowledge the HIV Transmission Elimination AMsterdam (H-TEAM) Consortium for facilitating the implementation of this project.

HIV Transmission Elimination AMsterdam (H-TEAM) Consortium:

Lead author:

G.J. de Bree (g.j.debree@amsterdamumc.nl).

H-TEAM members:

T. van Benthem[1], D. Bons[2], G.J. de Bree[3;4], P. Brokx[5], U. Davidovich[1;6], F. Deug[7], S.E. Geerlings[4], M. Heidenrijk[3], E. Hoornenborg[1], M. Prins[1;4], P. Reiss[3;5], A. van Sighem[8], M. van der Valk[4;8], J. de Wit[9], W. Zuilhof[7].

H-TEAM Project Management:

N. Schat[3], D. Smith[3].

H-TEAM additional collaborators:

M. van Agtmael[10], J. Ananworanich[11], D. Van de Beek[12], G.E.L. van den Berk[13], D. Bezemer[8], A. van Bijnen[7], J.P. Bil[1], W.L. Blok[12], S.J. Bogers[4], M. Bomers[10], A. Boyd[1;8], W. Brokking[14], D. Burger[15], K. Brinkman[13], N. Brinkman[13], M. de Bruin[16], S. Bruisten[1], L. Coyer[1], R. van Crevel[17], M. Dijkstra[1], Y.T. van Duijnhoven[1], A. van Eeden[14], L. Elsenburg[14], M.A.M. van den Elshout[1], E. Ersan[18], P. E.V. Felipa[1], T.B.H. Geijtenbeek[19], J. van Gool[1], A. Goorhuis[4], M. Groot[14], C.A. Hankins[3], A. Heijnen[20;21], M.M.J Hillebregt[8], M. Hommenga[1], J.W. Hovius[4], Y. Janssen[22], K. de Jong[1], V. Jongen[1], N.A. Kootstra[23], R.A. Koup[24], F.P. Kroon[25], T.J.W. van de Laar[26;27], F. Lauw[28], M.M. van Leeuwen[5], K. Lettinga[29], I. Linde[1], D.S.E. Loomans[1], I.M. van der Lubben[1], J.T. van der Meer[4], T. Mouhebati[7], B.J. Mulder[1], J. Mulder[30], F.J. Nellen[4], A. Nijsters[7], H. Nobel[4], E.L.M. Op de Coul[31], E. Peters[10], I.S. Peters[1], T. van der Poll[4], O. Ratmann[32], C. Rokx[33], M.F. Schim van der Loeff[1;34], W.E.M. Schouten[13], J. Schouten[1], J. Veenstra[29], A. Verbon[33], F. Verdult[5], J. de Vocht[10], H.J. de Vries[1;34;35], S. Vrouenraets[29], M. van Vugt[4], W.J. Wiersinga[4], F.W. Wit[4;6], L.R. Woittiez[4], S. Zaheri[8], P. Zantkuijl[7], A. Żakowicz[36], M.C. van Zelm[37], H.M.L. Zimmermann[1].

Affiliations:

1. Department of Infectious Diseases, Public Health Service of Amsterdam, Amsterdam, the Netherlands

2. Trans United Europe, Amsterdam, The Netherlands

3. Department of Global Health, Amsterdam UMC–location AMC, and Amsterdam Institute for Global Health and Development, Amsterdam, the Netherlands

4. Department of Internal Medicine, Division of Infectious Diseases, Amsterdam UMC–location AMC, Amsterdam, the Netherlands

5. Dutch Association of PLHIV, Amsterdam, the Netherlands

6. Department of Social Psychology, University of Amsterdam, Amsterdam, the Netherlands

7. Soa Aids Nederland, Amsterdam, the Netherlands

8. Stichting HIV Monitoring, Amsterdam, the Netherlands

9. Department of Interdisciplinary Social Science: Public Health, Utrecht University, Utrecht, the Netherlands

10. Department of Internal Medicine, Amsterdam UMC–location VUMC, Amsterdam, the Netherlands

11. US Military HIV Research Program and the Henry M. Jackson Foundation for the Advancement of Military Medicine, Bethesda, United States

12. Center of Infection and Immunity Amsterdam (CINIMA), Department of Neurology, Amsterdam UMC–location AMC, Amsterdam, the Netherlands

13. Department of internal medicine, OLVG–location East, Amsterdam, the Netherlands

14. DC Klinieken, Amsterdam, the Netherlands

15. Department of Pharmacy, Radboud University Nijmegen Medical Center, Nijmegen, the Netherlands

16. Aberdeen Health Psychology Group, Institute of Applied Health Sciences, University of Aberdeen, Aberdeen, United Kingdom

17. Department of Internal Medicine, Radboud University Nijmegen Medical Center, Nijmegen, the Netherlands

18. Department of General Practice, Amsterdam UMC–location AMC, University of Amsterdam, Amsterdam, the Netherlands

19. Laboratory of Experimental Immunology, Amsterdam UMC–location AMC Amsterdam, the Netherlands

20. Sexology Center Amsterdam, Amsterdam, the Netherlands

21. GP practice Heijnen & de Meij, Amsterdam, the Netherlands

22. Primary Care Amsterdam and Almere (Elaa), Amsterdam, the Netherlands

23. Laboratory for Viral Immune Pathogenesis, Amsterdam UMC–location AMC Amsterdam, the Netherlands

24. Immunology Laboratory, Vaccine Research Center, National Institute of Allergy and Infectious Diseases, National Institutes of Health, Rockville, Maryland, USA

25. Department of Infectious Diseases, Leiden University Medical Center, Leiden, the Netherlands

26. Department of Medical Microbiology, OLVG, Amsterdam, the Netherlands

27. Department of Donor Medicine Research, Laboratory of Blood-borne Infections, Sanquin Research, Amsterdam, the Netherlands

28. Department of Internal Medicine, Medical Center Jan van Goyen, Amsterdam, the Netherlands

29. Department of Internal Medicine, OLVG–location West, Amsterdam, the Netherlands

30. Department of Internal Medicine, Slotervaart Hospital (former), Amsterdam, the Netherlands

31. Epidemiology and Surveillance Unit, Center for Infectious Disease Control, National Institute of Public Health and the Environment, the Netherlands

32. School of Public Health, Faculty of Medicine, Imperial College London, London, United Kingdom

33. Department of Internal Medicine and Infectious Diseases, Erasmus Medical Center, Rotterdam, the Netherlands

34. Center for Infection and Immunology, Amsterdam (CINIMA), Amsterdam UMC–location AMC, University of Amsterdam, Amsterdam, the Netherlands

35. Department of Dermatology, Amsterdam UMC–location AMC, University of Amsterdam, Amsterdam, the Netherlands

36. AIDS Healthcare Foundation, Amsterdam, the Netherlands

37. Department of Virology, Erasmus Medical Center, Rotterdam, the Netherlands

## Author Contributions

**Conceptualization:** Saskia Bogers, Nynke van Dijk, Jan van Bergen.

**Data curation:** Saskia Bogers, Maarten Schim van der Loeff, Jan van Bergen.

**Formal analysis:** Saskia Bogers.

**Funding acquisition:** Suzanne Geerlings, Jan van Bergen.

**Investigation:** Saskia Bogers, Maarten Schim van der Loeff, Jan van Bergen.

**Methodology:** Saskia Bogers, Maarten Schim van der Loeff, Anders Boyd, Nynke van Dijk, Suzanne Geerlings, Jan van Bergen.

**Project administration:** Saskia Bogers, Jan van Bergen.

**Resources:** Nynke van Dijk, Suzanne Geerlings, Jan van Bergen.

**Supervision:** Maarten Schim van der Loeff, Anders Boyd, Suzanne Geerlings, Jan van Bergen.

**Validation:** Maarten Schim van der Loeff, Anders Boyd, Jan van Bergen.

**Visualization:** Saskia Bogers, Jan van Bergen.

**Writing – original draft:** Saskia Bogers.

**Writing – review & editing:** Saskia Bogers, Maarten Schim van der Loeff, Anders Boyd, Nynke van Dijk, Suzanne Geerlings, Jan van Bergen.

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
