## [Decision Letter · Decision Letter 0]

31 Jan 2023

PONE-D-22-35055Improving provider-initiated testing for HIV and other STI in the primary care setting in Amsterdam, the Netherlands: results from a multifaceted, educational intervention programmePLOS ONE

Dear authors,

Thank you for submitting your manuscript to PLOS ONE. After careful consideration, we feel that it has merit but does not fully meet PLOS ONE’s publication criteria as it currently stands. Therefore, we invite you to submit a revised version of the manuscript that addresses the points raised during the review process.

We look forward to receiving your revised manuscript.

Kind regards,

Guangming Zhong

Academic Editor

PLOS ONE

Journal Requirements:

2. One of the noted authors is a group or consortium HIV Transmission Elimination AMsterdam (H-TEAM) Consortium. In addition to naming the author group, please list the individual authors and affiliations within this group in the acknowledgments section of your manuscript. Please also indicate clearly a lead author for this group along with a contact email address.

Additional Editor Comments:

minor revision:

Please revise to address the minor concerns raised by the two reviewers.

Reviewers' comments:

Reviewer's Responses to Questions

**Comments to the Author**

1. Is the manuscript technically sound, and do the data support the conclusions?

Reviewer #1: Yes

Reviewer #2: Yes

2. Has the statistical analysis been performed appropriately and rigorously? 

Reviewer #1: Yes

Reviewer #2: Yes

3. Have the authors made all data underlying the findings in their manuscript fully available?

Reviewer #1: Yes

Reviewer #2: Yes

4. Is the manuscript presented in an intelligible fashion and written in standard English?

Reviewer #1: Yes

Reviewer #2: Yes

5. Review Comments to the Author

Reviewer #1: The manuscript submitted by Saskia Bogers et al presented an investigation into the effects of an educational intervention program on HIV and other types of sexually transmitted infections (STI) testing in Amsterdam, the Netherlands. Overall, the methods and results are sound. However, it still needs to go through several minor revisions before it can be published in our journal.

• Page 15 line 340: it seems that the authors only collected the data over the past 10 years rather than 20 years.

• How about the HIV and STI trend for testing ordered by GPs not attending the education program over the past 10 years as compared to what presented in this manuscript? Presenting this set of data could help to better understand the effects of the educational intervention program.

Reviewer #2: Author reported an educational intervention to improve HIV and STI testing in primary care

in Amsterdam, the Netherlands, which has certain clinical and social significance.There are no problems with the design, implementation and results of the test. I have some questions:

1. Why is the relative change of all STI pathogens generally increased in people aged 50-64?

2. Why did HIV testing peak appear in 2011 and fall to the lowest in 2014? What are the positiverates in 2011 and 2020?

3.Why did the HIV detection rate show a decline in 2020?

4.Line 296, please add “of” after the word “percentage”

Please explain it in the discussion.

6. PLOS authors have the option to publish the peer review history of their article (what does this mean?). If published, this will include your full peer review and any attached files.

Reviewer #1: **Yes: **Mian Huang

Reviewer #2: No

---

## [Author Response · Author response to Decision Letter 0]

6 Feb 2023

Reviewer #1: 

The manuscript submitted by Saskia Bogers et al presented an investigation into the effects of an educational intervention program on HIV and other types of sexually transmitted infections (STI) testing in Amsterdam, the Netherlands. Overall, the methods and results are sound. However, it still needs to go through several minor revisions before it can be published in our journal.

Reply: We thank the reviewer for their comments and their time to review our paper.

Page 15 line 340: it seems that the authors only collected the data over the past 10 years rather than 20 years.

Reply: We thank the reviewer for spotting this error and we have adjusted this accordingly:

From: “…as well as the collection of comprehensive data on HIV and STI testing by nearly all Amsterdam GPs over the past 20 years” (page 15, line 353-354),

To: “…as well as the collection of comprehensive data on HIV and STI testing by nearly all Amsterdam GPs over nearly ten years” (page 15, line 353-354).

How about the HIV and STI trend for testing ordered by GPs not attending the education program over the past 10 years as compared to what presented in this manuscript? Presenting this set of data could help to better understand the effects of the educational intervention program.

Reply: We thank the reviewer for this interesting suggestion. For Figure 1 in our manuscript, we were able to calculate overall trends in HIV testing by GPs in Amsterdam per 10,000 residents, as we estimated we had collected 90-95% of all tests ordered. The same figure could not be made by participant status (i.e., GPs before participation, after participation, or who never participated), as we do not know what the denominator would be for each participant group (i.e., how many patients are under care of GPs before participation, after participation, or who never participated, as the number of patients under care per GP varies), to calculate the HIV test rate per 10,000 persons per year by participant group. However, we do know the median number of HIV, chlamydia and gonorrhea tests ordered per GP per quarter-year, and their participant status for each quarter. 

We observed that these numbers were relatively stable over time in the study, and we agree that this is valuable information. Originally, in our paper, we did not describe trends in testing by GPs who never participated in the educational program. We have now, following this suggestion, added this data in the manuscript in the following passages: 

From 2011 to 2020, the median number of HIV tests ordered by GPs who never participated decreased from 5, IQR 2-11 to 4, IQR 2-8. (page 8, line 169-170)

From 2011 to 2020, the median number of chlamydia tests ordered by GPs who never participated remained stable from 16, IQR 8-27 to 16, IQR 8-28. (page 11, line 220-221)

From 2011 to 2020, the median number of gonorrhoea tests ordered by GPs who never participated decreased from 16, IQR 8-27 to 15, IQR 7-26. (page 11, line 249-250)

Additionally, in our Poisson-regression models, we compared the numbers of tests between (2) GPs after participation and (3) GPs who never participated in the intervention with GPs before participation (1; reference group) (see pag 6, line 115-120). This way, we took any trends among non-participants into account when estimating the effect of the intervention.

Reviewer #2: 

Author reported an educational intervention to improve HIV and STI testing in primary care

in Amsterdam, the Netherlands, which has certain clinical and social significance. There are no problems with the design, implementation and results of the test. 

Reply: We thank the reviewer for their comments and their time to review our paper.

I have some questions:

1. Why is the relative change of all STI pathogens generally increased in people aged 50-64?

Reply: We agree with the reviewer that this is a remarkable finding, as HIV and other STI are more often found in younger patients. However, In the Netherlands, the median age at HIV diagnosis is 40 years, and 30% of people newly diagnosed are 50 years or older, justifying proactive HIV testing of persons with indicator conditions and/or at risk in this age group, and likely explaining the increase in HIV testing in this group among GPs who participated in the educational programme. Likewise, there has been an increase in the percentage positive chlamydia and gonorrhoea tests among patients age 50 years and older over the last decade. We have now elaborated more about these findings in paragraph 2 of the Discussion:

In 2021, the median age at HIV diagnosis was 40 years, and 30% of people newly diagnosed with HIV were 50 years or older, justifying proactive HIV testing of persons with indicator conditions and/or at risk in this age group[2]. Likewise, an increase in the percentage positive chlamydia and gonorrhoea tests done at SHCs among patients age 50 years and older was observed over the last decade[7]. Conversely, as the median age at HIV diagnosis has been increasing over time[2], the decrease in HIV tests ordered among ≥65 year-old patients may lead to missed opportunities for HIV diagnosis if this group is inadequately tested in the future, although the number of persons diagnosed with HIV at ≥65 years is small in absolute numbers. (page 13, line 286-294)

2. Why did HIV testing peak appear in 2011 and fall to the lowest in 2014? What are the positiverates in 2011 and 2020?

Reply: One possible explanation for the decrease in HIV testing from 2011 to 2014 is a decrease in perceived HIV risk among both patients and GPs, as the incidence of HIV was steadily decreasing in the Netherlands over the same time period, due to measures effectively preventing HIV transmission. However, other developments happened in this same time-period including an increase of the Dutch compulsory annual deductible for patients from €170 to €385 (GP diagnostics are paid out-of-pocket by patients who have not yet exhausted their deductible), while meanwhile, HIV testing at STI clinics, which is free for the patient, increased. We have previously speculated about the changes in overall HIV testing, see reference 13 in this current paper, and the discussion on page 15, line 367-375.

As the reviewer suggested, we have now added information about the trend in HIV positivity over time:

From 2011 to 2020, overall HIV positivity declined from 0.8% to 0.5%. (page 10, line 201)

3. Why did the HIV detection rate show a decline in 2020?

Reply: We did perceive a decline in HIV testing from 2019 to the first quarter of 2020, as depicted in figure 1. One explanation for this observation is that there may have been a decrease in perceived HIV risk among patients and GPs, leading to less HIV testing, although other factors may have been at play, including financial barriers (patients will likely not have depleted their annual deductible in the first quarter of a year so that they have to pay for any HIV/STI testing out of pocket). To exclude the effect the COVID-19 pandemic had on healthcare delivery and therefore also on HIV testing, we already excluded Q2-4 in 2020 from our data. We have now added a reflection on this drop in testing and its significance for future initiatives, in the discussion:

We did observe a small decrease in overall testing by GPs in Amsterdam in the first quarter of 2020, which may reflect a decrease in perceived HIV risk among patients and GPs, among other factors. This finding is of particular importance in the context of a shrinking HIV epidemic, in which keeping GPs motivated for proactive HIV testing may be challenging in the future, when incidence and therefore perceived risk may decline. (page 14, line 316-320)

4. Line 296, please add “of” after the word “percentage”

Reply: We have adjusted this as suggested by the reviewer (page 13, line 308).

---

## [Decision Letter · Decision Letter 1]

20 Feb 2023

Improving provider-initiated testing for HIV and other STI in the primary care setting in Amsterdam, the Netherlands: results from a multifaceted, educational intervention programme

PONE-D-22-35055R1

Dear Dr. Bogers,

We’re pleased to inform you that your manuscript has been judged scientifically suitable for publication and will be formally accepted for publication once it meets all outstanding technical requirements.

Kind regards,

Guangming Zhong

Academic Editor

PLOS ONE

Additional Editor Comments (optional):

The authors have adequately addressed all concerns raised by the reviewers.

Reviewers' comments:

Reviewer's Responses to Questions

**Comments to the Author**

1. If the authors have adequately addressed your comments raised in a previous round of review and you feel that this manuscript is now acceptable for publication, you may indicate that here to bypass the “Comments to the Author” section, enter your conflict of interest statement in the “Confidential to Editor” section, and submit your "Accept" recommendation.

Reviewer #1: All comments have been addressed

Reviewer #2: All comments have been addressed

2. Is the manuscript technically sound, and do the data support the conclusions?

Reviewer #1: Yes

Reviewer #2: Yes

3. Has the statistical analysis been performed appropriately and rigorously? 

Reviewer #1: Yes

Reviewer #2: Yes

4. Have the authors made all data underlying the findings in their manuscript fully available?

Reviewer #1: Yes

Reviewer #2: Yes

5. Is the manuscript presented in an intelligible fashion and written in standard English?

Reviewer #1: Yes

Reviewer #2: Yes

6. Review Comments to the Author

Reviewer #1: The authors have addressed all my concerns and I have no objections for its publication in our journal.

Reviewer #2: The author has answered all my questions , which helped me better understand the background and results of the study.The revised manuscript is now acceptable for publication.

7. PLOS authors have the option to publish the peer review history of their article (what does this mean?). If published, this will include your full peer review and any attached files.

Reviewer #1: **Yes: **Mian Huang

Reviewer #2: No

---

## [Editor Report · Acceptance letter]

23 Feb 2023

PONE-D-22-35055R1 

Improving provider-initiated testing for HIV and other STI in the primary care setting in Amsterdam, the Netherlands: results from a multifaceted, educational intervention programme 

Dear Dr. Bogers:

I'm pleased to inform you that your manuscript has been deemed suitable for publication in PLOS ONE. Congratulations! Your manuscript is now with our production department. 

Kind regards, 

on behalf of

Dr. Guangming Zhong 

Academic Editor

PLOS ONE